# Hebb Alone Is Enough: Purely Excitatory Networks Self-Decorrelate to Expand Representation

## Abstract

We show that local, unsupervised Hebbian plasticity is sufficient for purely excitatory recurrent networks to self–decorrelate their population activity, thereby expanding representational dimensionality—without supervision. In a twin-reservoir protocol to isolate the causal effect of plasticity, and across both rate-based and spiking reservoirs driven by naturalistic audio (Japanese Vowels, Cats-Dogs), four canonical rules (Oja, BCM, pairwise STDP, triplet STDP) consistently reduce pairwise correlations, PCA-based metrics and spike-time synchrony relative to frozen controls, while maintaining stable dynamics in the echo-state regime. We provide a simple mechanistic account: when two neurons are strongly correlated, Hebbian plasticity pushes neurons to operate into distinct nonlinear regimes, decorrelating their outputs, lowering redundancy and yielding richer population codes. These results identify a minimal and biologically plausible route to high-dimensional coding and offer a hardware-friendly recipe for upgrading reservoir architectures with on-chip, unsupervised local plasticity. Our findings bridge machine learning and systems neuroscience by showing how Hebbian synapses alone can sculpt random recurrent substrates into high-capacity representational engines.

## 1 Introduction

Reservoir Computing (RC) and other semi-random neural substrates owe much of their power to the dimensionality expansion produced by fixed, randomly connected units. Yet both deep-learning practitioners and neuromorphic-hardware designers are converging on the same roadblock: random projections are task-agnostic, and biologically implausible.

Mixed selectivity, wherein individual neurons respond to combinations of multiple features, allows neural circuits to flexibly represent diverse stimuli and contexts. Expanding the dimensionality of input features enhances the separability of complex patterns, aiding tasks such as classification, recognition, and decision-making (Rigotti et al., 2013; Fusi et al., 2016).

A mechanistic account of how local, hardware-friendly Hebbian synapses can self-organize such reservoirs into high-dimensional representations (or how biological circuits achieve the same effect) remains unclear. Although plasticity rules have been tried in many reservoir-computing settings with only marginal benefits, clear theoretical motivation for their use has remained elusive.

### 1.1 Previous interpretations of Hebbian plasticity

Early theoretical work framed Hebbian plasticity as a local mechanism for unsupervised feature extraction in feedforward networks. The first of this kind, Oja's rule (Oja, 1982) showed that Hebbian plasticity could train a weight vector to converge to the first principal component of the input distribution. The Generalized Hebbian Algorithm of Sanger (1989) extended this logic to multi-unit layers, successively extracting orthogonal principal components by enforcing an output-decorrelation constraint.

In recurrent networks, Hebbian updates were first interpreted as a substrate for deterministic associative memory. Hopfield (1982) demonstrated that an outer-product Hebb rule sculpts an energy

landscape whose minima store binary patterns. Boltzmann machines (Hinton et al., 1984) (BMs) generalize this idea by replacing deterministic neurons with stochastic binary units and by using a combination of Hebbian learning in a phase with real inputs, and *anti-Hebbian* decay in a hallucinating phase. Training the weights so that the network's stationary distribution reproduces the statistics of the data which then encodes an entire probability landscape rather than a set of point attractors.

Finally, a body of work in Hebbian learning has simply tried to make sense of biological data and made models that mimic more closely the brain. The Bienenstock–Cooper–Munro (BCM) rule (Bienenstock et al., 1982) equips each synapse with a sliding threshold, dynamically balancing potentiation and depression so that only inputs driving activity above a homeostatic set point are strengthened. Pair-based spike-timing–dependent plasticity (STDP) formalized how millisecond-scale spike order biases synaptic change toward causally relevant inputs (Markram et al., 1997; Bi & Poo, 1998). Triplet and voltage-dependent extensions reproduced BCM-like thresholds in firing-rate regimes and embedded precise spike sequences in recurrent circuits (Pfister & Gerstner, 2006).

## 1.2 PLASTICITY IN RESERVOIR COMPUTING

Several lines of work have investigated how local synaptic plasticity can enhance reservoir computing, broadly falling into two categories: Hebbian-type potentiation (and its homeostatic extensions) and anti-Hebbian or decorrelation-focused updates.

On the Hebbian side, early attempts used biologically inspired rules to enrich the reservoir's dynamics. Yusoff et al. (2016) evaluated the classical BCM rule in echo state networks (ESNs) and reported modest, hard-to-reproduce gains. Building on this, Iacob et al. (2023) introduced a delay-sensitive variant of BCM that yields more consistent improvements on temporal benchmarks, while Cazalets & Dambre (2023) showed that a simple homeostatic rule with Hebbian potentiation can marginally boost forecasting accuracy. In parallel, spiking-network implementations by Meng et al. (2011) and Chrol-Cannon & Jin (2015) incorporated pair-based and triplet STDP, respectively, demonstrating timing-dependent potentiation. Finally, moving beyond purely software models, Lee et al. (2023) demonstrated that a phase-change photonic reservoir can self-tune its nonlinearity and memory via Hebbian-inspired rules, highlighting the promise of plasticity in hardware-based RC systems.

A complementary body of work has explored non-Hebbian rules. Babinec & Pospíchal (2007) pioneered the use of an "anti-Oja" update in ESNs, achieving lower prediction errors on the Mackey–Glass time series and subsequent studies by Morales et al. (2021) and Wang et al. (2021) combined anti-Oja's rule with intrinsic plasticity (IP) and/or heterogeneous neuron-specific parameterizations, finding that these hybrids deliver incremental but consistent performance gains. Meanwhile, Lazar (2009) applied binary on/off synaptic updates in a structurally adaptive reservoir, further improving pattern separation on custom benchmarks.

## 1.3 LINK BETWEEN RESERVOIR COMPUTING AND NEUROSCIENCE

Neuroscientific interest in reservoir computing comes from the observation that random, recurrently connected networks produce rich, high-dimensional population dynamics akin to those measured in cortex. The two fields often use different terms for closely related ideas: where machine learning speaks of high-dimensional projections or random features, systems neuroscience emphasizes *mixed selectivity*. Enel et al. (2016) argued that reservoir networks capture the highly recurrent nature of cortical microcircuits, and that the high-dimensional dynamics produced by random connectivity mirror the diverse mixtures observed in primate cortex.

The computational advantage of such high-dimensional representations is that they turn nonlinear input–output relationships into linearly separable ones (a consequence formalized by Cover's theorem; (Cover, 1965)). Rigotti et al. (2013) and Fusi et al. (2016) showed that neurons with nonlinear mixed selectivity increase the dimensionality of population codes, enabling simple linear classifiers to implement a large repertoire of functions (Rigotti et al., 2013; Fusi et al., 2016). Geometric analyses reveal that mixed-selective networks create high-dimensional polytopes in activity space; a linear readout can partition this space with a hyperplane, whereas low-dimensional codes cannot support the same flexibility (Barak et al., 2013; Tye et al., 2024). This is also analogous to the kernel trick in machine learning: by projecting inputs into a rich feature space, complex tasks become solvable with a linear classifier (Buonomano & Maass, 2009).

This reservoir perspective naturally separates representation and learning. Because the recurrent network already produces a rich variety of dynamical mixtures, learning rules need only adjust the synapses projecting to downstream targets. Indeed, models of context-dependent decisions in prefrontal cortex have shown that a fully recurrently connected network can perform flexible computations when only a single linear readout is trained (Mante et al., 2013; Enel et al., 2016). Likewise, experiments with cultured neuronal networks—treated as physical reservoirs—have achieved accurate classification of static inputs using linear decoders, emphasizing that only the readout weights need be learned while the internal reservoir remains fixed (Sumi et al., 2023).

### 1.4 OUR CONTRIBUTION

Our contribution is threefold :

- We propose a new interpretation of Hebbian plasticity that naturally decorrelate network activities. Intuitively (Fig. 1), when two neurons are strongly correlated, Hebbian potentiation strengthens the synapse between them. This increased connectivity drives their activities into different nonlinear regime of the activation function, effectively pushing their outputs apart and reducing redundancy which in turn promotes decorrelated, higher-dimensional population codes.
- We empirically validate this across settings. Using Oja, BCM, pairwise STDP, and triplet STDP, we observe (i) decrease pairwise correlations in rate-based networks, and (ii) reduce spike synchrony in leaky integrate-and-fire (LIF) reservoirs.
- This novel perspective that Hebbian plasticity act as decorrelation mechanisms in excitatory recurrent networks for time-series offers a simple unsupervised mechanism for observed mixed-selectivity but also a biologically plausible route to high-dimensional coding in reservoir computing.

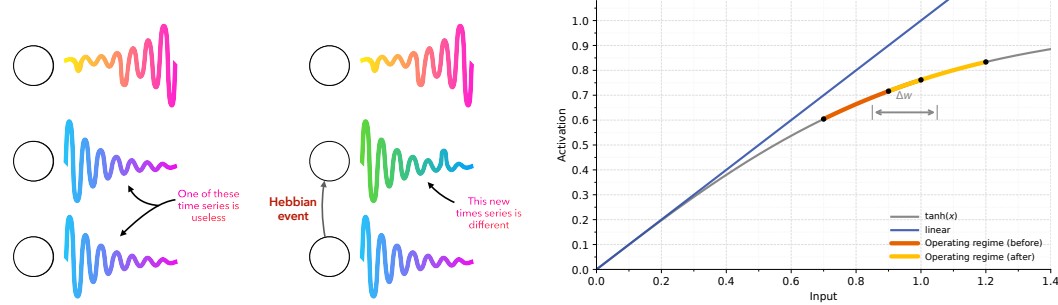

Figure 1: **Self-decorrelation via local Hebbian plasticity.** (left) Overview: initially redundant units diverge after a local Hebbian event, reducing pairwise correlations. (right) Mechanism: potentiation by $\Delta w$ shifts a unit's input along the saturating nonlinearity $\tanh$, moving its operating regime (orange→yellow). This unequal gain decorrelates activity.

The remainder of this paper is organized as follows. Section 2 introduces our methodological framework, including the reservoir models, Hebbian learning rules, datasets used and dimensionality metrics. Section 3 presents the main experiments, where we measure changes in correlation as plasticity is enabled, directly compare plastic and static twin reservoirs, and generalize our findings to spiking neurons. Finally, Section 4 summarizes our contributions and outlines directions for future research on plasticity-driven high-dimensional representations.

## 2 METHODS

### 2.1 NEURAL NETWORKS DYNAMICS: RATE-BASED AND SPIKE-BASED MODELS

We consider two classes of recurrent neural networks: a continuous-valued (rate-based) model and a spiking model based on leaky integrate-and-fire (LIF) dynamics. This dual-model approach enables

us to probe whether Hebbian decorrelation is a function of the underlying neural code (rate vs. spike), or whether it reflects a general principle of self-organizing recurrent networks. The LIF extension also provides stronger biological plausibility for our central claim: that local unsupervised plasticity yields dimensional expansion in biologically realistic settings.

**Rate-Based Reservoirs.** Rate neurons evolve according to standard echo state dynamics:

$$\mathbf{r}(t + 1) = \tanh\left(W\mathbf{r}(t) + s_{in} W_{\text{in}}\mathbf{x}(t) + s_{bias}\,\mathbf{b}\right), \tag{1}$$

where $\mathbf{r}(t) \in \mathbb{R}^N$ is the reservoir state, $W$ is the recurrent weight matrix, $W_{\text{in}}$ is the input projection, $\mathbf{x}(t)$ is the input vector, and $\mathbf{b}$ is a bias vector. The nonlinearity $\tanh$ ensures bounded activation. We tune the *input scaling* $s_{in}$ and *bias scaling* $s_{bias}$ to prevent saturation and ensure rich transient dynamics. Since $W$ will be adapted by the Hebbian rules, we do not explicitly optimize spectral radius in this paper.

**Spike-Based Reservoirs.** For biological realism, and to assess whether plasticity-induced decorrelation generalizes beyond rate dynamics, we also simulate a recurrent network of leaky integrate-and-fire (LIF) neurons. In rate-based reservoirs, neuron activity $r_j(t)$ in plasticity equations directly corresponds to the reservoir state $r_j(t)$. In spike-based reservoirs, neuron activities is represented by neuron's membrane potential $v_i(t)$ which evolves according to:

$$\tau_m \frac{dv_i(t)}{dt} = -v_i(t) + \sum_j W_{ij}s_j(t) + I_i(t), \tag{2}$$

where $\tau_m$ is the membrane time constant, $W_{ij}$ are synaptic weights, $s_j(t)$ is the spike train from presynaptic neuron $j$, and $I_i(t)$ is external input. When $v_i(t)$ crosses a threshold $\theta_{\text{spike}}$, the neuron emits a spike and $v_i(t)$ is reset.

We encode continuous input $\mathbf{x}(t)$ into deterministic spike trains using rate-to-spike transformations. We use a deterministic input encoding (as opposed to Poisson spike train for instance), in order to ensure that initial neuron populations exhibit shared input-driven correlations. This allows us to meaningfully test whether plasticity rules can decorrelate activity; otherwise, with independent random spike trains, no initial correlations would exist for plasticity to act upon.

## 2.2 Hebbian rules used

We endow the (excitatory-only) reservoir with *local, unsupervised* synaptic updates drawn from four canonical families and avoid the unstable plain Hebb rule due to unbounded growth. For rate networks we use (i) **Oja's rule** which add postsynaptic normalization that prevents weight blow-up and aligns weights with dominant input directions (Oja, 1982) and (ii) the **BCM rule** with a sliding activity threshold that balances potentiation and depression (Bienenstock et al., 1982). For spiking networks we use (iii) **pairwise STDP**, which depends on pre–post spike timing, and (iv) **triplet STDP**, which adds rate/frequency dependence and recovers BCM-like behavior in the rate limit (Pfister & Gerstner, 2006). All synapses are constrained to be non-negative with hard clipping, and hyperparameters are chosen to keep dynamics in the echo-state regime. Full update equations, parameter settings, and implementation details are provided in Appendix A. Empirically, these local rules are sufficient to drive decorrelation and dimensionality expansion without inhibition (Sec. 3).

## 2.3 Experimental setup

We evaluate on two realistic audio classification datasets **Japanese Vowels** (Mineichi Kudo, 1999) and **CatsDogs** (Thakoor & Gao, 2005). Japanese use convert *Mel-Frequency Cepstral Coefficients* (MFCCs) features using `librosa` (McFee et al., 2015). MFCCs are min–max normalized per coefficient to $[0, 1]$ to match our non-negative weights. (Full dataset descriptions are in Appendix C.)

To ensure a consistent reservoir size and equal treatment of all the dimension of the time series, we replicate each $d$-dimensional input vector $k$-times until the reservoir contains at least $N_{\min} = 200$ neurons, so $k = \lceil 200/d \rceil$. Thus, for the Japanese Vowels ($d = 12$) we obtain $12 \times 17 = 204$ units.

All recurrent weights $w_{ij}$ are initialized by sampling from a uniform distribution on $[0, 1]$, and we similarly enforce non-negativity on all input weights and biases. This choice serves only to emulate

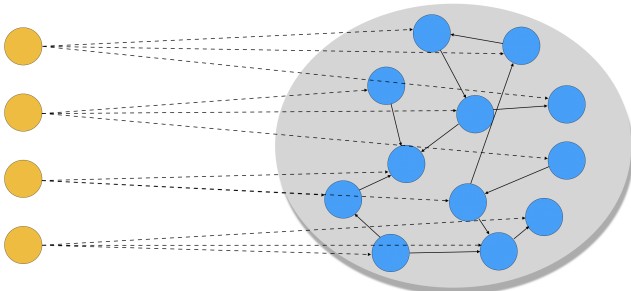

Figure 2: Schematic architecture of our setup. There are k time series after preprocessing (here 4). Each time series is fed to the same number of neurons (here 3). Each reservoir neuron receives exactly one of the input signals. The reservoir recombines the inputs

an "excitatory-only" reservoir in a minimal setting; it does not reflect the true balance of excitation and inhibition in biological neural circuits.

## 2.4 METRICS FOR EVALUATING DIMENSIONALITY EXPANSION

We assess decorrelation and feature–space growth with three complementary measures which complete description are given in Appendix B. For rate networks, we track the **mean pairwise Pearson correlation** over sliding windows and **PCA-based effective dimensionality**, defined as the smallest number of principal components required to explain 90% of the variance in the reservoir state matrix. For spiking networks, we use the **Spike Time Tiling Coefficient** (STTC) Cutts & Eglen (2014), which remains informative under sparse firing and avoids the floor effects of Pearson (see Appendix B, Figure 9).

## 3 RESULTS

### 3.1 HEBBIAN PLASTICITY WITH RATE-BASED NEURONS

**Regime Tuning**   Our objective is to evaluate the effect of Hebbian plasticity under stable, non-saturating dynamics. To this end, we choose a parameter grid that keeps healthy neurons dynamics. All reservoirs are initialized with positive weight matrix with connectivity of 0.1 and receive the entire concatenation of the datasets' sequences. Plasticity is disabled during the first 1,000 steps (warm-up).

As seen in Fig. 3, none of the sampled configurations drive the network into saturation, confirming that all models operate in a robust, analytically tractable regime. In Fig. 4 we can see that the sum of all excitatory weights for both Oja and BCM remains bounded under all tested hyperparameter settings. Finally we tracked the spectral radius $\rho(W)$ of each reservoir matrix over training to check whether the echo-state property is preserved. Figure 4 shows that across all datasets and both Oja and BCM rules, $\rho(W)$ remains in a healthy range (approximately 0.8–1.1), never pushing the network into unstable or saturated regimes, confirming that local Hebbian updates preserve the reservoir's dynamical stability.

**Evaluation of correlation**   With the same set of hyperparameters yielding good behaviors we generate purely excitatory random recurrent matrices and, after the warm-up, apply the Hebbian rule, recording the evolution of the linear correlation between reservoir states.

However, because the input itself also evolves over time, it remains ambiguous whether this decrease is driven primarily by the Hebbian updates or by natural fluctuations in the input statistics. To isolate the net contribution of synaptic plasticity to population decorrelation, we ran two identical reservoirs in parallel: a *static* copy whose recurrent matrix $W_{stat}$ remained fixed after initial randomization; a *plastic* copy whose matrix $W_{\text{plast}}$ was updated by either Oja or BCM rules after a 1000-step warm-up.

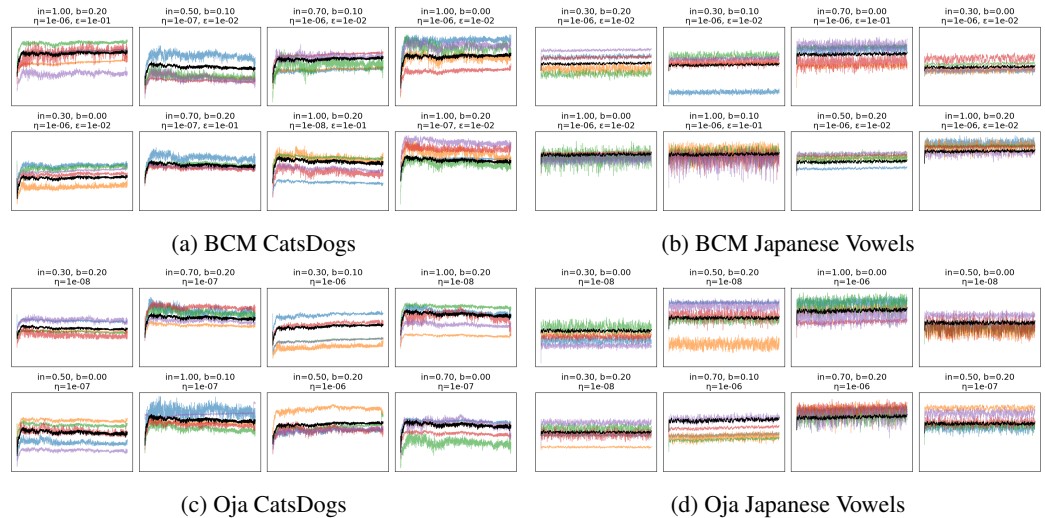

Figure 3: **Activation trajectories across for a sample of hyperparameter for 5 neurons** (colored lines). For readability, we display 8 representative configurations for each dataset/rule. Black line shows the population average for that configuration.

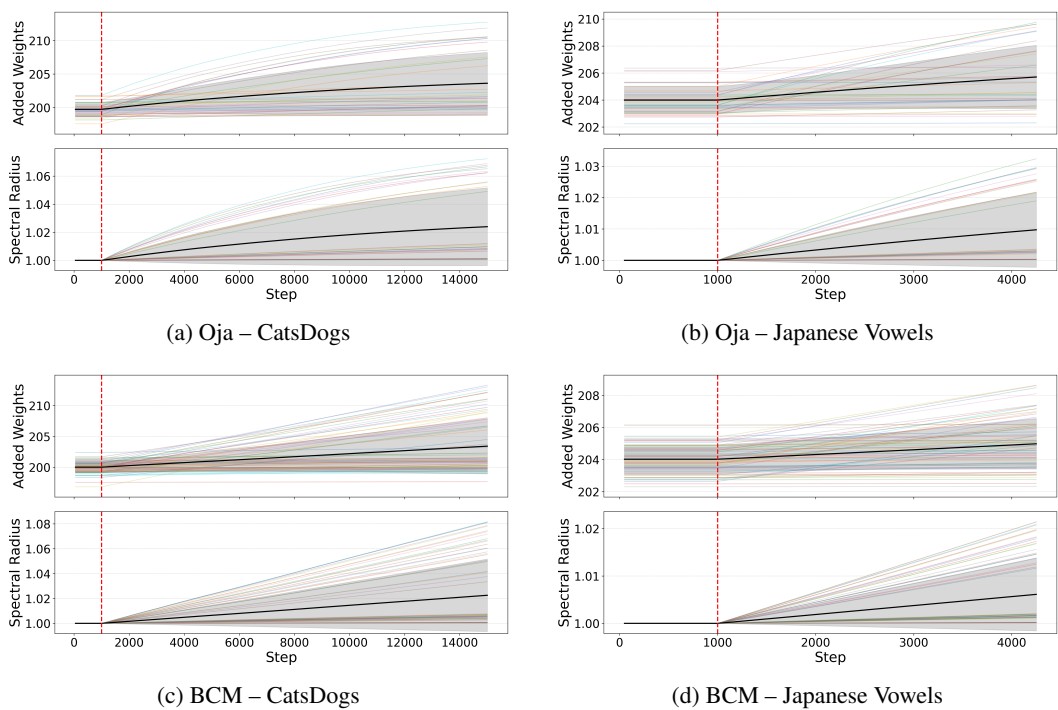

Figure 4: Top of each tile: total excitatory synaptic strength. Bottom: spectral radius $\rho(W)$. The vertical dashed red line marks the end of the 1 000-step warm-up and the start of plasticity.

Both copies received the *same* input stream and used the same input/bias weights, guaranteeing that any divergence in activity is attributable solely to synaptic updates. Every 50 steps we recorded the reservoir states, computed the mean pair-wise correlation $\rho_{\text{stat}}$ and $\rho_{\text{plast}}$, and logged their difference

$$\Delta\rho = \rho_{\text{plast}} - \rho_{\text{stat}} \ .$$

A negative $\Delta\rho$ therefore indicates that plasticity *de-correlated* the population relative to the frozen control, while a positive value indicates increased redundancy.

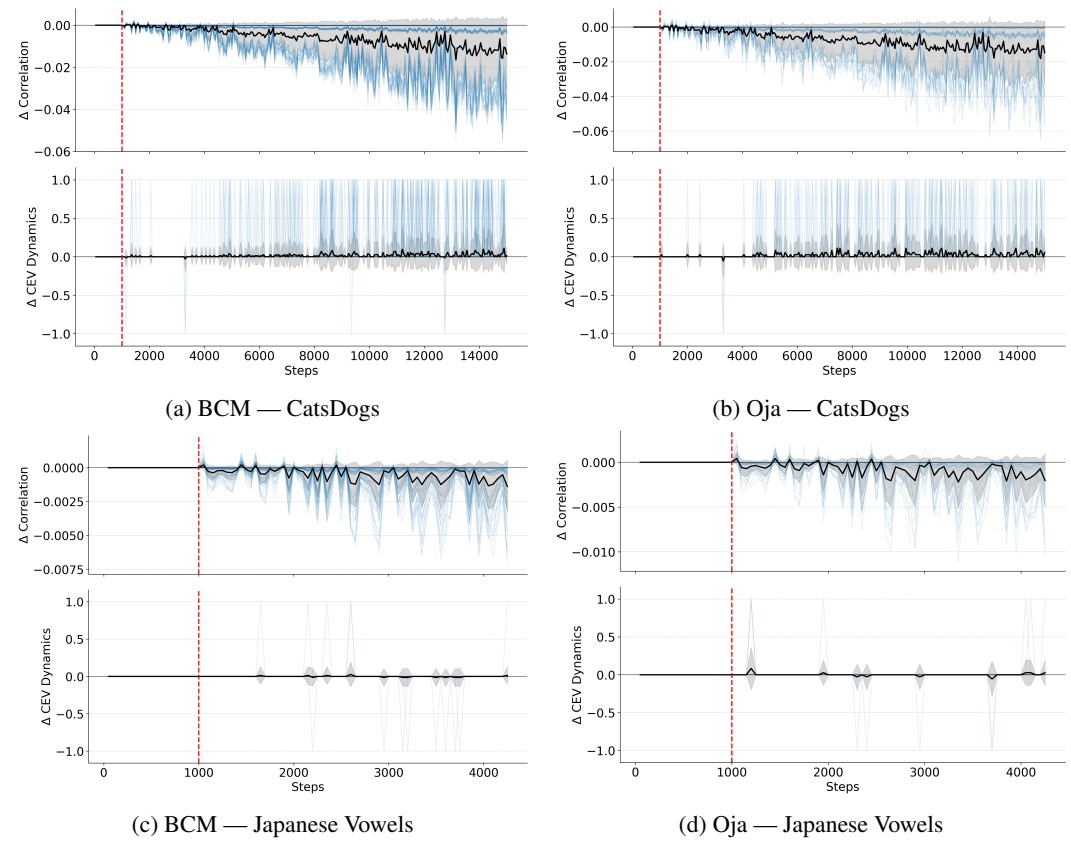

Figure 5: Top of each tile: $\Delta$ Pearson ($\rho_{\text{plast}} - \rho_{\text{stat}}$). Bottom: change in effective dimensionality $\Delta D = D_{\text{plast}} - D_{\text{stat}}$. Thin blue lines show individual runs; black is the population mean; gray is ±1 s.d. The vertical dashed red line marks the end of the 1 000-step warm-up and the start of plasticity.

Fig. 5 shows that on average $\Delta\rho$ steadily *below zero*. BCM exhibits a more pronounced positive overshoot after learning commences compared to Oja, but subsequently drifts downward; by the final third of training, the mean $\Delta\rho$ is negative. The large spread reveals a stronger sensitivity to $(\eta, \theta)$: some settings decorrelate quickly, whereas others remain near the noise ceiling for several thousand steps. The bottom panels show the sum of plastic weights converging toward stable values, confirming the overall convergence of the Hebbian plasticity rules.

We applied the same twin-reservoir protocol (static vs. plastic) to compute $D_{\text{stat}}$ and $D_{\text{plast}}$ every 50 steps for each hyperparameter setting. We then define

$$\Delta D = D_{\text{plast}} - D_{\text{stat}},$$

so that $\Delta D > 0$ indicates an increase in effective dimensionality due to plasticity. Figure 5 plots $\Delta D$ across all settings for both Oja and BCM on the CatsDogs and Japanese Vowels datasets. In every case, $\Delta D$ rises above zero immediately after the 1 000-step warm-up and plateaus at positive values, demonstrating that Hebbian plasticity consistently expands the reservoir's feature space. BCM typically yields slightly larger $\Delta D$ than Oja, with mean expansions of around 25 components on CatsDogs and 20 on Japanese Vowels.

### 3.2 HEBBIAN PLASTICITY WITH SPIKE-BASED NEURONS

To confirm that the dimensionality-expansion effects observed in our rate-based reservoirs generalize to biophysically more realistic system with leaky-integrate-and-fire (LIF) neurons driven by deterministic input encoders of the same two previous datasets. Crucially, we assessed population synchrony using the STTC instead of correlation as described in Section 2.

**Regime Tuning**   Similarly to our rate-based setting, we choose a set of parameters for our experiments that don't promote unhealthy regimes (saturation or total silence of the neurons).

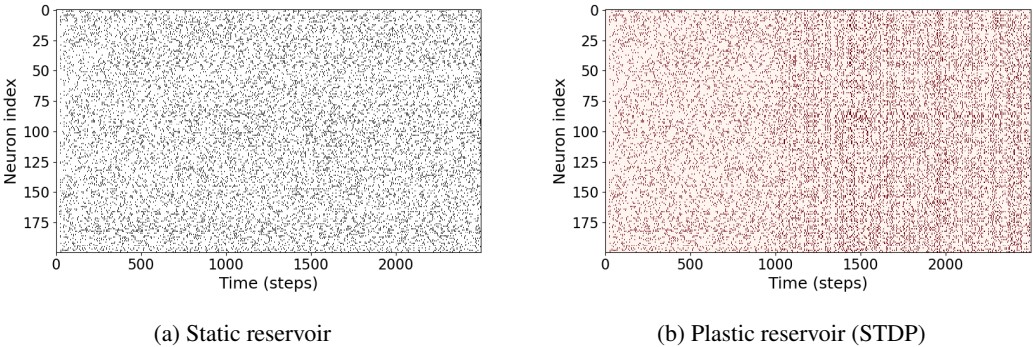

(a) Static reservoir                                          (b) Plastic reservoir (STDP)

Figure 6: Spiking activity in static vs. STDP-trained reservoir. Each panel shows binary spike rasters over the first 2 500 steps on the CatsDogs dataset.

**Evaluation of correlation**   We computed the mean pairwise STTC over 200-ms windows for both the static and plastic versions of the reservoir illustrated in Fig 6 and logged the difference:

$$\Delta\text{STTC} = \text{STTC}_{\text{plast}} - \text{STTC}_{\text{stat}} \tag{3}$$

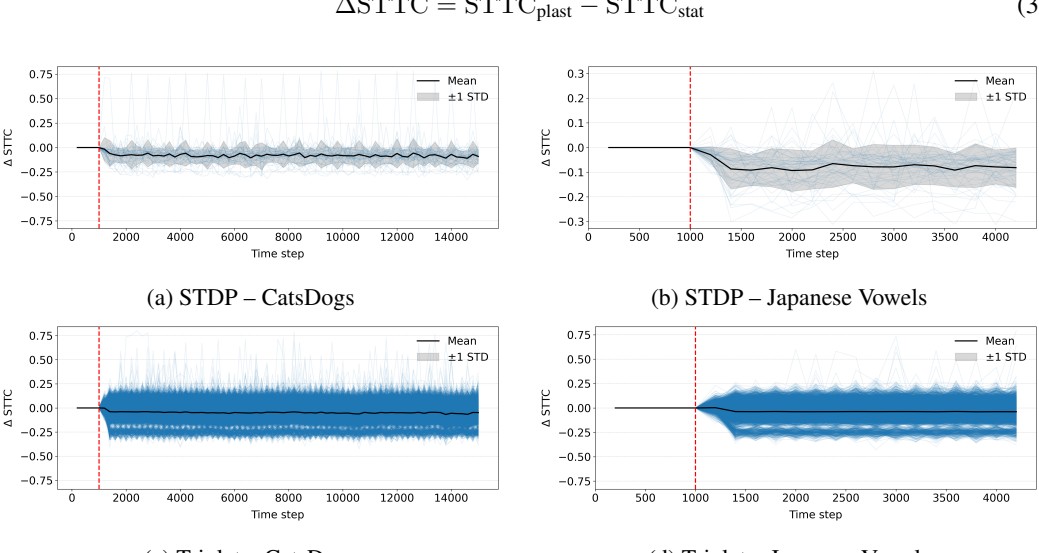

(a) STDP – CatsDogs                                          (b) STDP – Japanese Vowels

(c) Triplet – CatsDogs                                       (d) Triplet – Japanese Vowels

Figure 7: **Change in spike synchrony induced by plasticity.**   Curves show $\Delta\text{STTC} = \text{STTC}_{\text{plast}} - \text{STTC}_{\text{stat}}$; the grey band is $\pm1$ std across all hyperparameters. Plasticity consistently *reduces* pairwise synchrony, aligning with the decorrelation observed on rate traces.

A negative $\Delta\text{STTC}$ indicates that plasticity has desynchronized the population relative to the fixed control.

Fig 7 summarizes the trajectory of $\Delta\text{STTC}$ across all hyperparameter settings. Immediately after the warm-up phase, plastic reservoirs display *lower* STTC than their static counterparts, mirroring the decorrelation effect previously measured with correlation on rate traces. STDP rule produces a swift, stable reduction in spike synchrony, plateauing on average within approximately $2,000$ steps. The triplet rule is more contrasted but creates as well this .

### 3.3   DO REPRESENTATIONAL CHANGES TRANSLATE INTO PERFORMANCE GAINS?

We now ask whether the decorrelation and dimensionality expansion induced by local Hebbian plasticity improve downstream task performance. We compare reservoirs with static recurrent weights

to reservoirs with Oja or BCM plasticity, using a linear readout trained on *CatsDogs* and *Japanese Vowels*. Figure 8 shows the *change in mean test accuracy* (in percentage points, pp) relative to the static baseline for each dataset–rule pair (positive values favor plasticity).

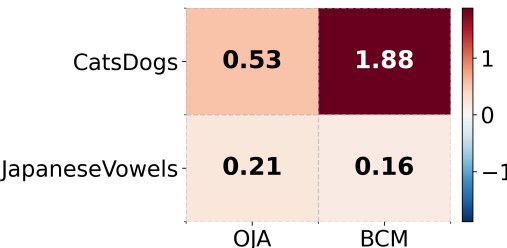

Figure 8: Improvement in mean test accuracy relative to a static reservoir (percentage points).

Plasticity yields consistently positive but *modest* gains: on *CatsDogs*, BCM shows the largest mean improvement ($\approx +1.88$ pp) and Oja is smaller ($\approx +0.53$ pp); on *Japanese Vowels*, both rules provide minor gains ($\approx +0.21$ and $+0.16$ pp for Oja and BCM). Overall, performance gains are small and dataset-dependent: plasticity never hurts here and sometimes helps, with the largest benefit on *CatsDogs*. This trend is broadly consistent with the stronger decorrelation observed for that dataset (cf. Fig. 5), suggesting that representational changes can translate into accuracy improvements, but the effect size depends on task structure and readout difficulty.

## 4 DISCUSSION

We investigated whether local, unsupervised Hebbian plasticity can reorganize purely excitatory recurrent reservoirs into higher–dimensional, less redundant representations, and whether these changes help downstream performance. Across two audio datasets and two modeling regimes (rate and spiking), four canonical rules (Oja, BCM, pairwise STDP, triplet STDP) showed consistently that plastic reservoirs decrease population correlations relative to frozen controls.

Do these representational changes help performance? With a simple linear readout, test-set gains were consistently positive but modest suggesting that plasticity-induced expansion can translate into accuracy improvements when task structure benefits from richer features.

Taken together, these results identify a minimal and hardware-friendly recipe for upgrading reservoir computing: add local Hebbian synapses. Without global feedback or supervised credit assignment, such rules reliably sculpt random recurrent substrates into stable, high-capacity representations that are at least competitive with static baselines and sometimes better. Conceptually, this aligns with the mixed-selectivity view of cortical computation: by reducing pairwise correlations while preserving nonlinear combinations of inputs, Hebbian updates rotate and spread activity so that single units become tuned to feature conjunctions, yielding a richer, more orthogonal basis for linear readouts.It also dovetails with recent theory that recasts Hebbian learning as local online whitening and variance maximization with competition (e.g. Oja's normalization, BCM's sliding threshold, triplet-STDP's BCM-like limit), effectively performing kernel expansion under simple constraints. In short, our data place mixed selectivity on a concrete mechanistic footing: simple Hebbian synapses are sufficient to carve randomly wired reservoirs into low-redundancy, high-dimensional codes that downstream learners can readily exploit.

**Limitations and future work.** Our study use only two datasets, a broader task suites and analysis (temporal reasoning, memory, and noisy settings) will clarify when gains are largest. Combining Hebbian updates with complementary homeostatic mechanisms (e.g., intrinsic plasticity or synaptic scaling), mixed excitation–inhibition, or sparsity constraints may further enhance robustness and energy efficiency. Finally, the excitatory-only, local nature of these rules makes them natural candidates for neuromorphic deployment; benchmarking on physical substrates is a promising next step.

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

# A   DETAILS OF HEBBIAN PLASTICITY RULES

The simplest form of Hebbian plasticity is $\Delta w_{ij} = \eta \cdot r_i \cdot r_j$, where $\Delta w_{ij}$ is the synaptic weight change from presynaptic neuron $i$ to postsynaptic neuron $j$, $r_i$ is the activity of the presynaptic neuron, $r_j$ is the activity of the postsynaptic neuron, and $\eta$ is the learning rate controlling the speed of adaptation. While this rule captures the core Hebbian idea "cells that fire together, wire together" we do not use it in that paper as it is unstable, since weights may grow without bound.

**Oja's rule**   To address this, Oja proposed a stabilized Hebbian rule Oja (1982) that incorporates a normalization mechanism to constrain synaptic strength. The update becomes:

$$\Delta w_{ij} = \eta \cdot r_j \cdot (r_i - r_j \cdot w_{ij}), \tag{4}$$

where the term $-r_j \cdot w_{ij}$ acts as a decay proportional to the current weight and the square of the postsynaptic activity. This prevents uncontrolled weight growth and ensures convergence to the first principal component of the input data when applied to linear neurons.

**BCM rule**   Another influential extension is the Bienenstock–Cooper–Munro (BCM) rule Bienenstock et al. (1982), which introduces an adaptive threshold that separates potentiation from depression. The rule is given by:

$$\Delta w_{ij} = \eta \cdot (r_j \cdot (r_j - \theta_M) \cdot r_i - \epsilon \cdot w_{ij}), \tag{5}$$

where $\eta$ is the learning rate, $\epsilon$ is a small weight-decay constant, $\theta_M$ represents a spatially averaged activity threshold, specifically, it corresponds to the expectation of the postsynaptic neuron activity $E(r)$ averaged over the input patterns

**STDP rule**   Hebbian principles also extend to biologically grounded spike-based models. In our implementation we use a discrete-time, trace-based pair-STDP that closely matches the continuous window $\Delta w_{ij} = A_+ e^{-\Delta t/\tau_+}$ for $\Delta t > 0$ (LTP) and $-A_- e^{\Delta t/\tau_-}$ for $\Delta t < 0$ (LTD). At each simulation step (of size $\mathrm{d}t$) we update per-synapse traces and weights as follows:

$$\mathrm{pre}_j(t) = \left(1 - \tfrac{\mathrm{d}t}{\tau_+}\right) \mathrm{pre}_j(t - \mathrm{d}t) \; + \; s_j(t), \tag{6}$$

$$\mathrm{post}_i(t) = \left(1 - \tfrac{\mathrm{d}t}{\tau_-}\right) \mathrm{post}_i(t - \mathrm{d}t) \; + \; s_i(t), \tag{7}$$

where $s_j(t), s_i(t) \in \{0, 1\}$ are the binary spike indicators of the presynaptic neuron $j$ and postsynaptic neuron $i$, respectively. The weight update is then

$$\Delta w_{ij}(t) = A_+ \, s_i(t) \, \mathrm{pre}_j(t) \; - \; A_- \, \mathrm{post}_i(t) \, s_j(t),$$

and we apply

$$W_{ij}(t) \; \longleftarrow \; \mathrm{clip}\big(W_{ij}(t) + \Delta w_{ij}(t), \, 0, \, w_{\max}\big).$$

Here $A_+$ and $A_-$ are the potentiation and depression amplitudes, $\tau_+$ and $\tau_-$ the corresponding time constants, and $w_{\max}$ the hard upper bound on excitatory synaptic strength. All weights are kept nonnegative and clipped at a maximum $w_{\max}$.

**Triplet STDP**   To capture both precise spike-timing effects and frequency-dependent plasticity (and to recover BCM-like behavior in the rate limit), we implement the Pfister & Gerstner (2006) triplet-STDP rule in its four-term form. Each synapse maintains four exponential spike-detector

variables:

$$r_1(t + \Delta t) = \left(1 - \frac{\Delta t}{\tau_+}\right) r_1(t) + s_j(t), \tag{8}$$

$$r_2(t + \Delta t) = \left(1 - \frac{\Delta t}{\tau_x}\right) r_2(t) + s_j(t), \tag{9}$$

$$o_1(t + \Delta t) = \left(1 - \frac{\Delta t}{\tau_-}\right) o_1(t) + s_i(t), \tag{10}$$

$$o_2(t + \Delta t) = \left(1 - \frac{\Delta t}{\tau_y}\right) o_2(t) + s_i(t), \tag{11}$$

where $s_j(t)$, $s_i(t) \in \{0, 1\}$ are the presynaptic and postsynaptic spike indicators, respectively.

Synaptic weight updates are then applied event-driven, with an infinitesimal causal shift $\epsilon \to 0^+$:

$$\text{at } t = t^{\text{pre}}: \quad w_{ij}(t) \longrightarrow w_{ij}(t) - o_1(t)\left[A_2^- + A_3^- r_2(t - \epsilon)\right], \tag{12}$$

$$\text{at } t = t^{\text{post}}: \quad w_{ij}(t) \longrightarrow w_{ij}(t) + r_1(t)\left[A_2^+ + A_3^+ o_2(t - \epsilon)\right]. \tag{13}$$

Here $A_2^{\pm}$ and $A_3^{\pm}$ set the amplitudes of pair-based and triplet-based potentiation/depression, respectively. All weights are kept nonnegative and clipped at a maximum $w_{\max}$.

# B   METRICS FOR EVALUATING DIMENSIONALITY EXPANSION

In this paper, we approximate dimensionality using correlation-based measures, choosing different metrics depending on whether the network is rate-based or spike-based:

**(i) Pairwise correlation $\rho$.**   For two neurons $i$ and $j$, the linear correlation coefficient, $r_{ij}$, measuring the linear relationship between their respective activation states, $x_i[t]$ and $x_j[t]$, over a time period $T$ is defined as:

$$r_{ij} = \frac{\sum_{t=1}^{T}(x_i[t] - \bar{x}_i)(x_j[t] - \bar{x}_j)}{\sqrt{\sum_{t=1}^{T}(x_i[t] - \bar{x}_i)^2}\sqrt{\sum_{t=1}^{T}(x_j[t] - \bar{x}_j)^2}}, \tag{14}$$

where:

- $x_i[t]$ and $x_j[t]$ are the activation states of neurons $i$ and $j$ at time $t$,
- $\bar{x}_i = \frac{1}{T}\sum_{t=1}^{T} x_i[t]$ and $\bar{x}_j = \frac{1}{T}\sum_{t=1}^{T} x_j[t]$ are their respective mean activations over the period $T$.

Over a window, the mean correlation between all distinct neuron pairs is :

$$\rho(t) = \frac{2}{N(N-1)} \sum_{i<j} r_{ij} . \tag{15}$$

Values of correlation range from $-1$ (anti-synchrony) to $+1$ (synchrony).

**(ii) Spike Time Tiling Coefficient.**   In sparse-spiking regimes, correlation becomes unreliable due to its sensitivity to long silent periods. To address this, we use the Spike Time Tiling Coefficient (STTC) Cutts & Eglen (2014), a spike-based synchrony measure that is robust to firing rate differences. As shown in Fig. 9, STTC captures input redundancy more reliably than correlation, making it a better probe of shared variability in low-activity regimes.

STTC measures the temporal clustering of spikes across trains, independent of firing rate. Given two spike trains $A = \{t_k^A\}_{k=1}^{N_A}$ and $B = \{t_\ell^B\}_{\ell=1}^{N_B}$, and a coincidence window $\Delta T$, we define:

- $L_A$, $L_B$: the total time within $\pm\Delta T$ of any spike in $A$ or $B$, respectively.
- $T_A$, $T_B$: the corresponding fractions of the full observation interval $T_{\text{win}}$,

$$T_A = \frac{L_A}{T_{\text{win}}}, \quad T_B = \frac{L_B}{T_{\text{win}}}.$$

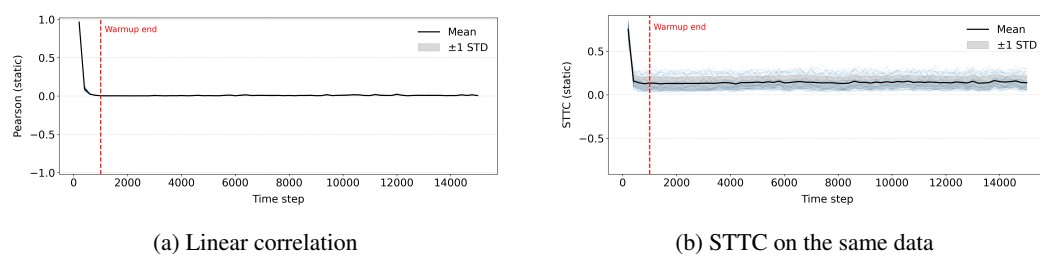

(a) Linear correlation          (b) STTC on the same data

Figure 9: Pairwise linear correlation and STTC recorded in a spiking reservoir. Across the spiking reservoir, correlation values collapse towards zero while STTC remains informative.

- $P_A$ and $P_B$: the fraction of spikes in $A$ (or B) that lie within $\pm\Delta T$ of at least one spike in $B$ (or A)

$$P_A = \frac{\left|\{\, k : \exists\, \ell,\ |t_k^A - t_\ell^B| \leq \Delta T \}\right|}{N_A}\,, \quad P_B = \frac{\left|\{\ell : \exists\, k,\ |t_\ell^B - t_k^A| \leq \Delta T \}\right|}{N_B}\,.$$

Finally, the Spike Time Tiling Coefficient is given by :

$$\text{STTC} = \frac{1}{2}\left(\frac{P_A - T_B}{1 - P_A\, T_B} + \frac{P_B - T_A}{1 - P_B\, T_A}\right). \tag{16}$$

Values of STTC range from $-1$ (perfect anti-synchrony) through $0$ (chance-level) to $+1$ (perfect synchrony), and remain meaningful even when spike trains are very sparse.

**(iii) Cumulative Explained Variance Dimensionality** Next we analyze the reservoir feature-space expansion. Intuitively, if the reservoir can represent data in a higher-dimensional space, then downstream linear classifiers or predictors should have an easier time separating different classes.

As an indication of the volume spanned by the reservoir states, we perform PCA on $\mathbf{H}$ (or equivalently on $\mathbf{H}^\top$) to obtain singular values $\sigma_1 \geq \sigma_2 \geq \cdots \geq \sigma_n$. Each $\sigma_j^2$ is proportional to the variance captured by the $j$-th principal component. The cumulative explained variance up to the $d$-th principal component is then given by:

$$C_d = \frac{\sum_{j=1}^{d}\sigma_j^2}{\sum_{k=1}^{n}\sigma_k^2}\,. \tag{17}$$

To assess the effective dimensionality of the reservoir's state space, we determine the minimum number of principal components required to reach a threshold $\theta = 0.9$ of cumulative explained variance:

$$D = \arg\min_d\left(C_d \geq \theta\right). \tag{18}$$

A higher $D$ indicates that more components are needed to capture the same variance, reflecting a more expansive coding.

## C  DATASETS

We use two audio classification datasets in our experiments:

**Japanese Vowels.** The **Japanese Vowels** dataset consists of recordings of a fixed sequence of Japanese vowels spoken by nine male speakers. Each utterance captures subtle variations in pitch, articulation, and timing, making the dataset a valuable testbed for speaker identification and temporal pattern recognition Mineichi Kudo (1999). The data are provided as MFCC time series of variable length and dimension, requiring no additional preprocessing.

**CatsDogs.** The **CatsDogs** dataset is the audio counterpart to the classic binary classification task. It comprises WAV files sampled at 16 kHz, containing 164 recordings of cats (1,323 seconds total) and 113 of dogs (598 seconds) Thakoor & Gao (2005). The clips span a range of naturalistic vocalizations and background noise conditions.

# D  PARAMETERS

**Rate-based reservoirs.**

- the input scaling factor $s_{in} \in \{0.3,\ 0.5,\ 0.7,\ 1.0\}$,
- the bias magnitude $s_{bias} \in \{0.0,\ 0.1,\ 0.2\}$,
- the learning rate $\eta \in \{10^{-6},\ 10^{-7},\ 10^{-8}\}$,
- for **BCM** only a weight decay term $\epsilon \in \{10^{-1},\ 10^{-2}\}$.

**Spike-based reservoirs (LIF).**  Unless noted otherwise, we reuse the same reservoir hyperparameters $(s_{in}, s_{bias})$ as above. Plasticity-specific sweeps are:

- **Pairwise STDP:** potentiation and depression amplitudes $A_+ \in \{0.005,\ 0.01\}$, $A_- \in \{0.006,\ 0.012\}$; time constants $\tau_+, \tau_-$ held fixed across runs (see App. A).
- **Triplet STDP:** pair/triplet amplitudes $A_2^+ \in \{0.005,\ 0.01\}$, $A_3^+ \in \{0.002,\ 0.005\}$, $A_2^- \in \{0.006,\ 0.012\}$, $A_3^- \in \{0.003,\ 0.006\}$; time constants $\tau_+ \in \{20,\ 40\}$ ms, $\tau_x \in \{40,\ 80\}$ ms, $\tau_- \in \{20,\ 40\}$ ms, $\tau_y \in \{40,\ 80\}$ ms.

