# OpenReview forum: "Hebb Alone Is Enough: Purely Excitatory Networks Self-Decorrelate to Expand Representation"
_ICLR.cc/2026/Conference — ICLR 2026 Conference Withdrawn Submission_

### Official Review · Reviewer_rzEA · 2025-10-30

**Soundness:** 1
**Presentation:** 2
**Contribution:** 1
**Rating:** 2
**Confidence:** 3

**Summary:**

This paper proposes that local, unsupervised Hebbian plasticity alone is sufficient for purely excitatory recurrent networks to achieve self-decorrelation of neuronal activity, thereby expanding the network's representational dimensionality, without external supervision or inhibitory connections.The authors test this claim across four canonical Hebbian/Hebe-like rules (Oja, BCM, pairwise STDP, triplet STDP) in both rate-based Echo State Networks (ESNs) and spiking Leaky Integrate-and-Fire (LIF) networks. Using a "twin-reservoir protocol" to isolate the effect of plasticity, they empirically demonstrate that the plastic networks reduce pairwise correlations and increase effective dimensionality.The core mechanism proposed is that Hebbian potentiation, driven by high correlations, pushes the targeted neurons into the non-linear saturation region of their activation function (e.g., $\tanh$). This saturation leads to an asymmetric reduction in gain, effectively decorrelating the output of highly redundant neurons and expanding the overall coding space.

**Strengths:**

1. Comprehensive Review of Plasticity in RC: The paper provides a thorough and well-organized review of existing literature, particularly focusing on the role of plasticity within the Reservoir Computing (RC) framework.

2. Novel Mechanism Hypothesis: The paper puts forth an original and counter-intuitive hypothesis regarding how excitatory Hebbian potentiation, coupled with non-linear saturation, could potentially lead to decorrelation.

**Weaknesses:**

1. Critical Lack of Theoretical Foundation (Major Flaw): This is the paper's most significant weakness. The central claim that Hebbian potentiation leads to decorrelation is based purely on an unsupported physical intuition without any mathematical proof or rigorous analytic derivation. This failure to provide a theoretical basis makes the entire mechanism logically unsound and unconvincing, especially since the mechanism contradicts the fundamental purpose of Hebbian learning (to increase correlation).

2. Flawed Logical Intuition and Mechanism Attribution: The paper fails to provide a convincing logical explanation for how Hebbian potentiation itself drives decorrelation. The claim that "Hebb Alone Is Enough" is highly questionable, as the rules used (Oja, BCM) include stabilization/normalization terms which are essential to prevent saturation. The observed effect is likely due to the non-Hebbian stabilizing components interacting with the saturation, making the paper's core attribution ambiguous and logically fragile. Furthermore, the reliance on saturation makes the mechanism non-generalizable to common non-saturating functions like ReLU.

3. Limited Empirical Validation: The experiments are restricted to simple ESN/LIF models and two small audio classification tasks (Japanese Vowels, CatsDogs). This limited scope, combined with the lack of theoretical grounding, is insufficient to demonstrate the mechanism's robustness, versatility, or general applicability to complex recurrent network settings.

4. Absence of Comparative Performance Analysis: The paper fails to compare the performance gain (readout accuracy) achieved by this Hebbian self-decorrelation against established optimization or plasticity methods in Reservoir Computing. This omission leaves the practical significance, efficiency, and overall scientific value of the proposed mechanism completely unclear.

**Questions:**

The following three points represent the most critical issues that must be addressed before this paper can be considered for acceptance.

1. Mechanism Attribution and Generalizability (Addressing Weaknesses 1 & 2):
The paper's central finding—that Hebbian learning leads to decorrelation—is counter-intuitive and currently lacks a rigorous theoretical foundation. We require the authors to clarify the exact source of the decorrelation effect and its dependence on the activation function.
**Mechanistic Disambiguation**: The rules employed (Oja, BCM) include stabilizing terms (e.g., normalization, dynamic threshold). Is the decorrelation primarily caused by (a) the core Hebbian potentiation itself, (b) the stabilization/normalization terms, or (c) the nonlinear coupling with the $\tanh$ saturation? Please provide a more formal, even if simplified analysis to explicitly demonstrate how the potentiation term leads to the minimization of correlation.
**Generalizability Test**: The current mechanism critically relies on the saturation property of the $\tanh$ function. We require an ablation study where the experiments are repeated using a non-saturating activation function, such as ReLU or GeLU, to demonstrate whether the claimed mechanism holds true beyond the specific choice of $\tanh$.

2. Experimental Complexity and Relevance (Addressing Weakness 3):
The current empirical validation is limited to simple recurrent network models and two small audio datasets. The contribution must be demonstrated on more challenging, real-world temporal tasks to establish the relevance of the mechanism to modern AI. Please apply the proposed Hebbian learning mechanism to a significantly more complex temporal processing task, such such as video sequence processing (e.g., action recognition) or a larger-scale time-series prediction task, to validate its effectiveness and scalability beyond simple audio classification.

3. Practical Significance and Comparative Analysis (Addressing Weakness 4):
Given that the theoretical contribution to neuroscience remains ambiguous, the paper's value must be established by demonstrating improved performance in an AI context. **Comparative Performance**: We require a crucial comparative analysis of the gain in readout accuracy. Please report the performance of your Hebbian-optimized reservoir against other established methods in plastic Reservoir Computing on the tested datasets. Without this comparison, the practical significance and value of the proposed "Hebb Alone Is Enough" mechanism remain completely unclear.

---

### Official Review · Reviewer_C9ss · 2025-10-31

**Soundness:** 2
**Presentation:** 3
**Contribution:** 2
**Rating:** 2
**Confidence:** 5

**Summary:**

**Summary**:
The authors of this paper use a simulation approach to empirically demonstrate the effect of a set of Hebbian learning rules on the characteristics of population activity/firing in excitatory recurrent networks, a.k.a echo-state networks or liquid state machines.

**Strengths:**

**Pros**:
- The main, and only point this paper is making is that Hebbian learning decorrelate population activity.
- Decorrelated population activities in recurrent networks lead to expanded high-dimensional representations, which (as empirically demonstrated) provide some modest performance gain which such representations are further processed by trainable learning layers.

**Weaknesses:**

**Cons**:
- The notion of Hebbian learning and its application to reservoir computing has been studied over many years, and there exist a lot of earlier results.
- This work does not present any major new insight other than an empirical observation that Hebbian learning leads to decorrelation, something not supervising at all.
- The key point made in the paper doesn't provide major advancement in the field of reservoir computing, and is expected to only have a very minor impact in real world applications.

**Questions:**

- The authors are encouraged to demonstrate a more indepth theoretical analysis of the decorrelation mechanism of Hebbian learning to provide more insights.

- In addition, how to further optimize Hebbian learning and/or the resulting decorrelation to provide a large performance gain should be discussed and demonstrated.

---

### Official Review · Reviewer_h6Ma · 2025-11-01

**Soundness:** 2
**Presentation:** 2
**Contribution:** 1
**Rating:** 4
**Confidence:** 3

**Summary:**

This paper studies whether pure excitatory recurrent reservoirs can, by themselves, become less redundant and more high-dimensional when equipped only with local, unsupervised Hebbian-like plasticity. The authors use a twin-reservoir protocol (static vs. plastic) on two audio time-series datasets (Japanese Vowels, Cats–Dogs) and four canonical rules (Oja, BCM, pairwise STDP, triplet STDP). They show that, after a warm-up, the plastic reservoir consistently exhibits (i) lower mean pairwise correlation, (ii) higher PCA-based effective dimensionality, while (iii) remaining in a stable echo-state regime. They further show small gains for a linear readout.

**Strengths:**

**Twin-reservoir protocol**: running static and plastic copies on exactly the same input stream is a clean way to causally attribute the decorrelation to plasticity rather than input drift.

**Breadth of rules & codes**: showing the same qualitative effect for Multiple rules, and for both rate and LIF reservoirs, makes the claim look like a principle.

**Stability check**: tracking total excitatory weight and spectral radius to address the stability concern.

**Weaknesses:**

**Effect size is modest**: downstream gains are small, which makes the practical ML impact less compelling.

**Excitatory-only but no E/I comparison**: since real circuits use inhibition for decorrelation, a minimal inhibitory baseline would help calibrate the benefit of “Hebb alone.”

**Lack of theoretical support**: the paper offers an intuitive “nonlinear-regime separation” story (Hebbian potentiation pushes correlated units into different parts of tanh), but there is no formal analysis showing that Oja/BCM/STDP in a recurrent, nonnegative network should yield $\Delta \rho <0$ or $\Delta D>0$; nor is there a low-dimensional dynamical or mean-field model backing the mechanism.

**Questions:**

If you take an E/I recurrent network (i.e. with an explicit inhibitory population or inhibitory plasticity) and use vanilla Hebb (without the postsynaptic decay term), do you get a decorrelation/expansion effect comparable to your E-only network with Oja/BCM? In other words, to what extent is the “postsynaptic suppression” built into Oja/BCM functionally substituting for what inhibition would do? A small controlled experiment here would make the “Hebb alone” claim much sharper.

---

### Official Review · Reviewer_PPjX · 2025-11-01

**Soundness:** 2
**Presentation:** 2
**Contribution:** 2
**Rating:** 2
**Confidence:** 3

**Summary:**

The paper studies the question of whether purely hebbian plasticity (in the forms of  Oja, BCM, pairwise STDP, triplet STDP) suffices to make a purely excitatory recurrent network decor relate its activity and increase the dimensionality od representations, required for classification, without any supervisory signal or inhibition.  The authors demonstrate both on rate based and spiking networks the effects of the Hebbian rules on two audio datasets as a decorrelation of the activity of the recurrent reservoir. They compare the results  to frozen controls using twin-reservoir setup, showing reductions in pairwise correlations/synchrony and small improvements in downstream classification accuracy.

I find the central claim of the paper a bit overstated relative to the evidence, and the numerical experiments carefully chosen to demonstrate the argued decor relation. For such a strong argument I would expect some theoretical/analytical justification.

**Strengths:**

- The paper presents a clear and testable central hypothesis: Hebbian plasticity alone, in the absence of inhibition or anti-correlating mechanisms, can drive decorrelation and expand representational dimensionality in recurrent excitatory networks.
- The work considers both rate-based and STDP plasticity.

**Weaknesses:**

- The argumentation refers only to idealised scenarios of a purely excitatory reservoir and for specific operating points.
- In a purely excitatory networks Hebbian co-activity typically increases representational similarity/corrrelations, not decreases it. The paper’s mechanism relies entirely on differential saturation of the nonlinearity or the operating point of the neurons sitting past the inflection point of the activation function near saturation. This is only argued qualitatively, not sufficiently demonstratetly quantitatively. Also I would guess that once both  neurons reach the saturation point, their activities will correlate again.
- The authors  replicate each input feature $k$ times so that “each time series is fed to the same number of neurons.” This input replication creates an easy decor relation target, since it creates clusters of neurons with same input and thus initially very high correlation, and any subsequent heterogeneity in the network results in reducing the correlation. So part of the reported effect may be an artifact of this very particular input configuration, not an intrinsic property of purely Hebbian plasticity in excitatory RNNs.
- For such a strong conceptual statement, there is no theoretical justification /derivation, that would not be difficult to obtain given the idelised setting.

**Questions:**

- Doesn’t the argument about the decorrelation due to change of operating point hold only in the absence of bi-directional connections/non-mutually connected neurons?
- How do you expect the argumentation to transfer in a more biologically realistic setting of networks with additional inhibition? Inhibition would also lead to a decrease of the operating point on the activation function pushing it into the linear regime that would have the opposite effects of the decor relation you are arguing about.
- Would it be possible to replace the Hebbian updates with random updates and report $\Delta \rho$ and $\Delta D$? This would provide a stronger evidence on whether the decor relation can be attributed to the Hebbian updates.
- You say spectral radius stays $0.8 - 1.1$. Does decorrelation still occur if you fix $\rho(W)$ by rescaling W after every update?
- For the STDP experiments, since you clip weights to $[0, w_{max}]$, how many synapses are actually saturated at the end?
- Also see weaknesses.

---

### Note · Authors · 2026-04-01

I have read and agree with the venue's withdrawal policy on behalf of myself and my co-authors.

---

### Meta-Review · Area_Chair_5fyN · 2026-01-04

**Summary:**

The reviewers expressed concern that the results are not sufficiently supported, which is especially important given that the result is counter-intuitive. Specifically, there was a lack of theoretical support, there were no comparisons against other mechanisms of decorrelation, and there may be potential statistical artifacts that are driving the results and need to be controlled for.

**Reviewer Concerns:**

The authors did not provide a rebuttal, so none of the concerns were addressed.

**Reviewer Scores:**

The authors did not provide a rebuttal, so I do not think that the reviewers would have changed their scores.

---

### Decision · Program_Chairs · 2026-01-26

Reject